# Antiplatelet Therapy in Low-Platelet-Count Patients After Percutaneous Coronary Intervention for Acute Coronary Syndromes

**DOI:** 10.3390/jcm14030838

**Published:** 2025-01-27

**Authors:** Francesco Paciullo, Paolo Gresele

**Affiliations:** 1Faculty of Medicine and Surgery, Vita-Salute San Raffaele University, Via Olgettina 58, 20132 Milan, Italy; 2Department of Medicine and Surgery, University of Perugia, Strada Vicinale Via Delle Corse, 06132 Perugia, Italy; paolo.gresele@unipg.it

**Keywords:** antiplatelet therapy, DAPT, gastrointestinal bleeding, intracranial hemorrhage, platelet transfusions

## Abstract

The risk of cardiovascular events increases considerably after an acute coronary syndrome (ACS), particularly in the first few months. Dual antiplatelet therapy represents the mainstay of secondary prevention during this period, but is associated with a not-negligible risk of bleeding which, among other factors, is influenced by the platelet count. Thrombocytopenic patients may experience an ACS, and several patients with ACSs develop thrombocytopenia during hospitalization: the management of antithrombotic therapy in this setting represents a challenge. Here, we review the available evidence on the use of antithrombotic therapy in patients with low platelet counts after an ACS.

## 1. Introduction

In recent decades, interventional cardiology has significantly improved the survival of patients with coronary disease. Stent technology has greatly progressed, from bare metal stents to ultrathin third-generation drug eluting stents with better polymer biocompatibility, contributing to amelioration of the outcomes of acute coronary syndromes (ACSs) [1,2]. However, stent thrombosis still represents a feared and life-threatening complication of stent implantation, with an incidence ranging from 0.5 to 2% [3]. Currently, dual antiplatelet therapy (DAPT), consisting of the combination of aspirin with a new-generation P2Y12 inhibitor, is considered the most efficacious therapeutic strategy to prevent stent occlusion and subsequent ischemic events [4]; accordingly, a minimum of 1 to 6 months of a DAPT course, based on the type of stent and coronary disease, is recommended to all patients undergoing stent implantation [5]. Even in patients with a chronic coronary syndrome (CCS) who have undergone PCI, a six-month course of DAPT is recommended. In ACS, if possible, DAPT should be continued for at least 12 months, extending to up to 2 years in some cases, using a reduced dose of ticagrelor, particularly in patients with anterior STEMI with a high ischemic risk and low bleeding risk [6]. Unfortunately, DAPT increases the overall incidence of bleeding, which in turn may require premature antiplatelet interruption, with a consequent rise in major adverse cardiovascular events (MACEs) [7]. For this reason, early de-escalation of dual therapy for both ACSs and CCSs should be reserved only for patients with a high or very high bleeding risk and a low thrombotic risk.

Platelets, circulating cell fragments derived from bone marrow megakaryocytes, are primary players in hemostasis. They are involved in the whole process of clot formation, enabling the arrest of bleeding after a vascular injury. Upon endothelial damage, platelets adhere to the exposed subendothelial collagen, aggregate and stimulate fibrin formation. Therefore, a defect in platelet count or function can be associated with a bleeding tendency of variable severity [8], influencing the net clinical benefit of antithrombotic therapy. Nevertheless, a low platelet count does not protect against arterial thrombosis [9], and thrombocytopenia is not rarely observed among patients with ACS, in whom it seems to predict both bleeding and ischemic events [10,11]. The management of thrombocytopenic patients undergoing coronary stent implantation thus represents a challenge, and requires a careful approach focused on the accurate evaluation of the individual’s bleeding and thrombotic risk [12]. In this review, we will discuss the available evidence and the key points for the management of patients with a low platelet count who are undergoing coronary revascularization.

## 2. Antiplatelet Agents in Acute Coronary Syndromes

Almost 50 million people are on antiplatelet therapy for primary and secondary prevention of cardiovascular disease (CVD) in the USA [13]. Antiplatelet drugs include a wide range of agents that blunt platelet activity by different mechanisms of action and have different impacts on the risk of bleeding (Table 1) [11]. Acetylsalicylic acid (ASA), or aspirin, is the most widely prescribed antiplatelet agent. It has been estimated to be prescribed in 24% of the general population and 32% of >70-year-old diabetics [14]. A more recent study reported that the prevalence of ASA use in primary prevention among people aged 60 years old or more, from 2012 to 2021, reached 29.7% in the US [15]. ASA targets the enzyme cyclooxygenase (COX)-1, thus inhibiting the synthesis of platelet thromboxane A2, a potent platelet agonist, and consequently platelet aggregation [16,17,18]. The second most important class of antiplatelet drugs is inhibitors of the platelet ADP receptor P2Y12 (clopidogrel, ticagrelor, cangrelor and prasugrel). The thienopyridines clopidogrel and prasugrel are irreversible inhibitors of P2Y12 which act as prodrugs, while ticagrerol and cangrelor (the latter administered intravenously) are directly active molecules which block the receptor competitively and reversibly [16]. GPIIb/IIIa receptor antagonists, including abiciximab, eptifibatide and tirofiban, which are all administered intravenously [19], are used in the setting of percutaneous coronary intervention (PCI) (infused immediately before, during and after the procedure) in association with heparin and ASA, or in the preoperative setting in patients who need surgery early after stent implantation [20].

The mainstay of pharmacologic treatment in ACS patients undergoing stent implantation is represented by DAPT, which is intended to reduce early ACS recurrence and stent occlusion. Although DAPT significantly lowers the risk of CV event recurrence after an ACS, it is associated with a not-negligible increase in bleeding [4]. In particular, among the P2Y12 inhibitors, ticagrelor and prasugrel have shown a higher efficacy profile compared to clopidogrel [23], at the expense of a worse safety profile for bleeding, limiting their use in patients prone to developing hemorrhages [24]. Unfortunately, currently, a large proportion of patients experiencing ACS have both a high bleeding and high thrombotic risk (bi-risk patients), and their management remains a challenge [25].

## 3. Hemorrhagic Risk of Antiplatelet Therapy

Antiplatelet therapy may be associated with a variably increased risk of bleeding, according to the dose, intensity, type of drug and number of antithrombotic agents (Table 1). In a recent metanalysis comparing the hemorrhagic risk of direct oral anticoagulants (DOACs) and of antiplatelet agents, the incidence of major hemorrhages among antiplatelet users reached 1.76% over a mean follow-up of 15.5 months, while fatal hemorrhages occurred in 0.28% of patients over a mean follow-up of 14.8 months, and the incidence of intracranial bleedings was 0.48% during a mean follow-up of 17.1 months [21]. While the association between antiplatelet treatment and bleeding from the gastrointestinal tract is well recognized, with a reported two-fold increased risk compared to a placebo (95% CI: 1.61–2.66) [26], its role in the occurrence of intracranial hemorrhage (ICH) is still debated [27]. Indeed, although previous studies on the use of ASA in primary and secondary prevention have not found an association with the occurrence of ICH [28], more recently, a significant 34% increased risk of intracranial bleeding with the use of aspirin compared to a placebo was reported [29]. Moreover, antiplatelet agents increase the risk of cerebral microbleeds, which in turn increase the risk of ICH [30], and ongoing antiplatelet therapy appears to be deleterious for outcomes in patients experiencing an ICH, independently from the cause (e.g., trauma, hypertensive crisis) [31]. Indeed, up to 30% of ICHs occur in subjects treated with antiplatelet agents, and in these patients, the risk of hematoma expansion is higher than in those who have not been treated with antiplatelet agents [27,32]. The combination of a P2Y12 inhibitor with ASA further increases the risk of bleeding. In particular, for 300 days following DAPT introduction, a 6% increased risk of hemorrhages (one-third of which are gastrointestinal) has been reported [33], with a 0.2–0.3% annual risk of ICH [27]. The bleeding risk of DAPT is more pronounced with prasugrel and ticagrelor compared to clopidogrel [17,34].

## 4. Bleeding Risk of Thrombocytopenia

Given the central role of platelets in hemostasis, it is expected that patients with a reduced platelet count are more prone to spontaneous and post-traumatic hemorrhages [35,36]. Although data on the bleeding risk of patients undergoing DAPT based on their platelet count is lacking, a count of at least 100 × 10^9^/L is an inclusion criterion in most clinical trials evaluating antithrombotic therapy after PCI [37]. According to the National Cancer Institute’s Common Terminology Criteria for Adverse Events (NCI’s CTCAE), recently modified by Falanga et al., thrombocytopenia severity is classified according to five grades (Table 2) [12]. Unfortunately, the grade of thrombocytopenia alone does not linearly predict individual bleeding risk [38], and very low platelet counts may not be associated with a proportionally increased incidence of hemorrhages [39]. This discrepancy is in large part due to the fact that bleeding tendency results from complex interactions among all the components of the hemostatic system, and that a normal count is not necessarily associated with a normal platelet function [38,40]. Based on these premises, it is very important to consider all the factors that are potentially associated with bleeding, including age, comorbidities, polypharmacy, bleeding history, frailty, and renal and hepatic function, without limiting the evaluation to the platelet count in patients with thrombocytopenia. In this context, age represents an element to be considered in the assessment of the individual bleeding risk in patients that are candidates to PCI for a myocardial infarction [41]; patients >70 years old experience a 7.5% incidence of in-hospital bleeding after PCI [42]. Accordingly, age is considered in almost all bleeding scores [42]. Moreover, the prevalence of thrombocytopenia increases with age [43,44]. Nevertheless, patients with coronary diseases, old age alone does not represent an absolute contraindication to DAPT [45]. In these patients, bleeding risk may be mitigated by using a radial approach to PCI and preferring clopidogrel over new-generation P2Y12 inhibitors [46]. In this regard, a recent meta-analysis, including 29,217 >75-year-old patients undergoing DAPT for an ACS, reported an 83% reduction in the incidence of bleeding events with clopidogrel vs. prasugrel or ticagrelor, with no differences in efficacy [47].

## 5. Thrombocytopenia in Patients with Acute Coronary Syndromes

Coronary revascularization with stent placement represents the mainstay of the management of critical coronary stenosis [48]. According to the clinical setting and the type of stent placed, PCI needs to be followed by DAPT to prevent thrombosis, for at least 1 month (with new-generation drug eluting stents) [49]. Due to the bleeding risk associated with DAPT, the decision to implant a stent should take into consideration the individual’s bleeding risk, including their platelet count [50]. Upon hospital admission, approximately 5% of patients with an ACS have thrombocytopenia, and 2.4 to 9.2% develop thrombocytopenia after PCI [51,52]. In the setting of ACSs, thrombocytopenia may be the consequence of an underling pre-existent disease, such as hyporegenerative or immune thrombocytopenia, or may represent a warning bell for prothrombotic conditions such as thrombotic thrombocytopenic purpura or antiphospholipid syndrome. Moreover, a drop in platelet count can result as a complication of medical treatment, such as heparin-induced thrombocytopenia (HIT) or GPIIb/IIIa-associated thrombocytopenia. In all situations, thrombocytopenia represents a negative prognostic predictor, and seems to identify a subset of patients with a greater comorbidity burden [53].

Interestingly, a pooled analysis from two clinical trials, including 10,603 patients with an ACS (ST-elevation or non-ST-elevation) showed a significant association between mild thrombocytopenia at hospitalization and mortality, and adverse cardiac events, with no impact on major or minor bleeding occurrence [54]. Data from the PURSUIT trial, evaluating 10,948 patients presenting with a non-ST-elevation (NSTEMI), reported an incidence of thrombocytopenia of 7%, with an associated increase in both bleeding and thrombotic events [55]. Among the 36,182 patients from the “Can Rapid Risk Stratification of Unstable Angina Patients Suppress Adverse Outcomes With Early Implementation” (CRUSADE) registry of the American College of Cardiology/American Heart Association, mild to severe thrombocytopenia significantly predicted the risk of in-hospital mortality and bleeding. In particular, mild thrombocytopenia (platelet count between 100–149 × 10^9^/L) was associated with an increased risk of mortality (adjusted odds ratio [OR], 2.01; 95% CI, 1.69 to 2.38) and bleeding (adjusted OR, 3.76; 95% CI, 3.43 to 4.12), and for each 10% drop in platelet count, there was a 40% increase in mortality risk [95% CI, 1.33 to 1.46] and a 90% increase in bleeding risk [95% CI, 1.83 to 1.95] [56]. Data from a pooled, patient-level analysis of the CHAMPION (Cangrelor Versus Standard Therapy to Achieve Optimal Management of Platelet Inhibition) trials, which compared cangrelor with clopidogrel for the prevention of thrombotic complications during and after PCI, showed that the development of thrombocytopenia, determined as a platelet count < 100 × 10^9^/L or a platelet drop of at least 50%, was associated with a 14-fold increase in major bleeding events and a 3-fold increase in major cardiovascular events [57].

In a recent analysis of the prospective GRACE registry, including 52,647 patients with an ACS, 152 (0.3%) were reported to develop HIT, 324 (0.6%) developed GPIIb/IIIa receptor antagonist-associated thrombocytopenia (defined as a platelet count of <100 × 10^9^/L occurring during hospitalization, after admission, in any patient who had received a glycoprotein IIb/IIIa inhibitor), and 368 (0.7%) developed other types of thrombocytopenia. Independently from the cause, these patients were significantly more likely to experience major bleeding, (re)infarction, or stroke [10]. Among the 7435 NSTE-ACS patients enrolled in the SYNERGY trial, 675 patients (9.1%) developed mild thrombocytopenia, and 139 patients (1.9%) developed severe thrombocytopenia. A low platelet count significantly increased the risk of bleeding and reduced the likelihood of discharge on guideline-recommended antiplatelet therapy; this may have contributed to these patients’ higher long-term mortality [58]. The impact of low platelet count at admission on cardiovascular outcomes during an ACS has been recently evaluated among patients enrolled in the Italian START ANTIPLATELET registry. Among patients admitted to coronary care units with an ACS, the presence of a low platelet count at admission was associated with an increased risk of major adverse clinical events (MACEs). In particular, MACE-free survival was significantly shorter in thrombocytopenic patients, particularly in those with a platelet count lower than 100 × 10^9^/L. Moreover, a low platelet count was an independent predictor of a MACE, together with age, atrial fibrillation, a low glomerular filtration rate and a low ejection fraction [59]. Moreover, another recent study on 1.198 patients with STEMI reported that a platelet count upon admission between 102 and 206 × 10^9/^L (vs. a count between 206 and 259 × 10^9^/L) was associated with 3-fold increase in 1-year all-cause mortality [60]. All of these data reflect the complexity of the management of antiplatelet therapy in patients with both thrombocytopenia and an ACS, and, in particular, the prevention of recurrence whilst avoiding bleeding events.

## 6. Thrombosis in Immune Thrombocytopenia

Immune thrombocytopenia (ITP) represents one of the most common acquired causes of thrombocytopenia. Interestingly, this condition is associated with both bleeding and thrombosis. Indeed, patients affected by ITP seem to experience more arterial and venous thrombotic events (VTEs) than control patients. The incidence of arterial events in patients with ITP reaches 2.6% per year [61]. In patients with ITP, the risk of thrombosis increases according to age and cardiovascular risk factors. Moreover, it may also be associated with treatment, such as splenectomy and TPO mimetics [62], and with the production of prothrombotic ultralarge von Willebrand factor multimers by megakaryocytes [63]. Unfortunately, this thrombotic risk is not counterbalanced by a reduction in bleeding occurrence. Patients with a mild to moderate platelet count reduction who have experienced a previous thrombotic event may be treated with antiplatelet therapy, tailoring the doses according to the platelet count and, in particular, using conventional therapy (including DAPT, when required), in the case of patients with a platelet count between 150 and 100 × 10^9^; while, in the case of patients with a platelet count between 50 and 100 × 10^9^, DAPT may be employed, but using clopidogrel as P2Y12 inhibitor. Patients with a platelet count between 25 and 50 × 10^9^ should only be treated with single antiplatelet therapy, while in the case of severe thrombocytopenia, treatment should be focused first on raising the platelet count to at least >25 × 10^9^L, using corticosteroids and/or immunoglobulins, before starting antithrombotics [64].

## 7. Thrombotic Thrombocytopenic Purpura in ACSs

Thrombotic thrombocytopenic purpura (TTP) is a life-threatening syndrome caused by a congenital or acquired reduction in the activity of the enzyme ADAMTS-13, which is involved in the cleavage of von Willebrand factor. TTP is characterized by microvascular thrombosis, platelet count reduction and hemolysis. Its general management includes plasma exchange, corticosteroids, rituximab and the use of caplacizumab, an anti-von Willebrand factor antibody that mitigates thrombosis. Overall, in 5–15% of cases, TTP presents as myocardial infarction [65,66]. This association may produce a mortality rate of 38–50% that is in part related to thrombocytopenia and its impact on therapeutic choices. In particular, the low platelet count and increased bleeding risk associated with caplacizumab can make it prohibitive the use of aggressive antiplatelet therapy, and consequently coronary stent placement [66,67,68]. Unfortunately, TTP can also be triggered by the use of P2Y12 inhibitors, mainly ticlopidine, but also clopidogrel, prasugrel and ticagrelor [68,69,70]. The management of ACS in the context of TTP may represent a challenge [68]; while single antiplatelet therapy is usually considered on top of general management, the choice of PCI and consequently dual antiplatelet therapy should be carefully evaluated on a case-by-case basis, taking into consideration the potential bleeding risk associated with thrombocytopenia and caplacizumab [71].

## 8. Heparin-Induced Thrombocytopenia

Heparin-induced thrombocytopenia (HIT) is a life-threatening condition associated with platelet consumption due to the action of anti-PF4 antibodies elicited by heparin. It is characterized by thrombosis and thrombocytopenia, without evidence of hemolysis or fibrinogen consumption. Anti PF4 antibodies are observed in 8.7% of ACS patients, while HIT in this context has an incidence of 1.6%, and is favored by the mechanical stress of PCI, a longer duration of heparin treatment, the type of heparin, a history of thrombosis and the presence of hyperlipidemia [72]. The management of HIT includes the prompt discontinuation of heparin and the initiation of a non-heparin anticoagulant, and the use of antiplatelet therapy should be closely weighted against bleeding risk. In the case of stent placement, dual antiplatelet therapy could be temporarily discontinued [73] in accordance with platelet count, after the initiation of a parenteral anticoagulant alternative to heparin.

## 9. Thrombocytopenia in Antiphospholipid Syndrome

Antiphospholipid syndrome (APS) is an autoimmune condition associated with arterial and venous thrombosis and pregnancy complications. In APS, thrombosis is sustained by the production of prothrombotic antibodies, activating coagulation, platelets and the endothelium [74,75]. In APS, myocardial infarction is observed in 2.5–5% of cases, and represents the most common cause of death [76]. Patients with APS often develop thrombocytopenia, with a reported incidence of 26% in patients with a platelet count < 100 × 10^9^/L, as a result of immune destruction or peripheral consumption [77]. Interestingly, the presence of thrombocytopenia is associated with a 3-fold increased risk of thrombosis [78], but it does not influence the incidence of bleeding, and it is more common among patients with systemic lupus erythematous (SLE) [79]. APS-associated thrombocytopenia may be treated with hydroxycloroquine in the case of SLE, and with immunosuppressors in the case of catastrophic APS. Due to the complex pathogenesis of APS, in patients who develop arterial thrombosis, antiplatelet therapy alone is not enough, and full-dose anticoagulation (including vitamin K inhibitors) associated with antiplatelet agents should be used in the case of ACS patients with stent placement. Unfortunately, there is no evidence to guide the use of antithrombotic therapy in subjects with a low platelet count. In this regard, a recent consensus of experts discouraged the use of both antiplatelet and anticoagulant agents in the case of severe thrombocytopenia [80]. In the absence of data on ACS patients with APS and thrombocytopenia, the use of antithrombotic therapy should follow the approach of other thrombotic thrombocytopenic conditions, and should be closely monitored. In these patients in particular, immunomodulatory therapy may help to attain platelet counts that are sufficient for full antithrombotic treatment [81].

## 10. Use of Dual Antiplatelet Therapy in Patients at High Bleeding Risk

Antiplatelet agents increase bleeding risk, especially when administered in combination. In particular, compared to single antiplatelet therapy, DAPT is associated with a 60% increase in bleeding risk [82]. This effect is particularly emphasized in the presence of an underlying bleeding tendency. Given that the bleeding risk following PCI correlates with the intensity of antithrombotic therapy, its duration and the number of antithrombotic agents [83], possible actions to mitigate bleeding risk include switching to a less potent antithrombotic drug (de-escalation) or downgrading from newer P2Y12 antagonists to clopidogrel and reducing DAPT duration [84]. However, a premature interruption of DAPT may be associated with insufficient antithrombotic protection and a consequent increase in cardiovascular recurrence. However, when physician-guided, discontinuation of DAPT appears to be safe [7,37]. In this regard, ESC guidelines recommend (class II b, level B) shortening the duration of DAPT to one month in all high-bleeding-risk patients (HBR) [6], with ≥1 major (including moderate-severe thrombocytopenia) or ≥2 minor criteria from the Academic Research Consortium [85]. In this context, the use of newer-generation stents, like polymer-free drug-coated stents (DCSs), which are associated with a lower rate of late thrombosis and thus require a shorter course of DAPT, may represent an option [86,87,88]. A recent meta-analysis of 15 clinical trials, including 35,326 patients with ACS, found no difference in MACE incidence when comparing a 1-month vs. 12-month duration of DAPT, with a significantly lower incidence of bleeding in the first group [89]. In the randomized MASTER DAPT trial on high-bleeding-risk patients who had undergone biodegradable-polymer sirolimus-eluting coronary stent implantation, no differences were reported in net adverse clinical events and major adverse cardiac or cerebral events between one month vs. three months of DAPT, with a significant reduction in bleeding events observed among patients treated with abbreviated therapy [90]. These data were confirmed by a subsequent meta-analysis, including 11 trials, evaluating DAPT in 9006 high-bleeding-risk patients, which showed that a 1- or 3-month abbreviated DAPT regimen was associated with lower bleeding and cardiovascular mortality, without increasing ischemic events, compared with a ≥6-month DAPT regimen [48]. However, thrombocytopenic patients were poorly represented in all these populations, and a sub-analysis in this group was not performed; indeed, in the group of patients with conditions associated with an increased risk of bleeding, less than 1% of patients with a low platelet count were included [91].

## 11. Management of Dual Antiplatelet Therapy in Patients with Thrombocytopenia

Although it is conceivable that a low platelet count may significantly impair hemostasis during DAPT, thrombocytopenia is not among the parameters considered in either the DAPT or in the PRECISE DAPT score, the two most frequently used tools to guide decisions regarding the long-term management of antiplatelet therapy after an ACS. Moreover, in most studies evaluating the management of ACS, patients with thrombocytopenia have been excluded or under-represented [34,92]. However, a meta-analysis including more than 110,000 patients from eight studies reported an increase in post-procedural total bleeding (OR: 1.89, 95% CI: 1.16–3.07; *p* = 0.01), access site bleeding (OR: 1.66, 95% CI: 1.15–2.39; *p* = 0.006), intracranial bleeding (OR: 1.78, 95% CI: 1.30–2.43; *p* = 0.0003), gastrointestinal bleeding (OR: 1.44, 95% CI: 1.14–1.82; *p* = 0.002) and any major bleeding (OR:1.67, 95% CI: 1.42–1.97; *p* = 0.00001) in thrombocytopenic patients treated with DAPT, compared with non-thrombocytopenic patients. Interestingly, the rate of overall total stent thrombosis (OR: 1.18, 95% CI: 0.90–1.55; *p* = 0.24), including definite and probable stent thrombosis, was not significantly different between the groups [93].

To date, there is no clinical evidence supporting a standardized management regime for antiplatelet therapy in ACS patients with thrombocytopenia, due to a lack of data; therefore, management is, to a large extent, based on expert opinion (Table 3). As a general rule, thrombocytopenic patients should avoid non-steroidal anti-inflammatory drugs, and should be treated with proton pump inhibitors to reduce the risk of gastrointestinal bleeding. All cardiovascular risk factors should be aggressively managed. Stent implantation, and thus DAPT, should be carefully considered in patients who are not at a very high cardiovascular risk (like patients with STEMI, early symptom onset, life-threatening arrhythmias or hemodynamic instability, and proximal coronary stenosis), and should preferably be performed when intravascular imaging is available [94], ensuring the use of adequately-sized stents and optimal stent implantation to lower the possible need of reintervention and reduce the required perioperative heparin dose. New-generation stents should be preferred to allow the shortest possible course of DAPT [95]. Antiplatelet therapy should be completely avoided in the case of a platelet count < 10 × 10^9^/L. The leading cause of thrombocytopenia should be always investigated and treated to raise the patient’s platelet count and allow for the use of conventional treatment. Accordingly, in patients with ITP with moderate to severe thrombocytopenia, treatment should include immunoglobulin therapy and/or steroids [64]. TPO mimetics should be carefully considered due to the potential deleterious effects in terms of thrombotic recurrence [96]. In patients with malignancy-related thrombocytopenia, platelet transfusions should be considered to raise the platelet count to >10 × 10^9^/L in order to allow the use of low-dose aspirin [12]. DAPT should be avoided in case of a platelet count < 50 × 10^9^/L, a history of bleeding, an anticipated further drop in platelet count or lack of recovery within 4 weeks following the planned intervention (e.g., ongoing bleeding, sepsis, accelerated consumption, splenic sequestration, graft vs. host disease) [97], distal coronary stenosis or stenosis of a side branch, complex coronary anatomy or coronary stenosis without an immediate impact on the clinical course. In these patients, conservative management should be considered [98], especially in patients who have experienced symptoms for >12 h and who have no signs of ongoing ischemia [84]. Patients at a higher thrombotic and bleeding risk may also be treated with plain old balloon angioplasty without stent placement [99]. In patients with no history of bleeding and a stable platelet count > 50 × 10^9^, dual antiplatelet therapy with aspirin and clopidogrel for six months can be used, while ticagrelor and prasugrel are not recommended [12,64]; finally, conventional treatments can be considered when the patient’s platelet count is >100 × 10^9^/L. The approach for thrombocytopenic patients with ACS is summarized in Table 4.

## 12. Potential Role of Hemostasis Tests in Optimizing Individual Bleeding Risk

A potential resource to guide antithrombotic therapy administration in thrombocytopenic patients may be hemostasis monitoring. Currently, many tests are available for evaluating platelet function, and they have been recently suggested for the assessment of the bleeding risk of patients on antiplatelet treatment who are candidates for surgery [100,101]. However, the role of hemostasis monitoring in managing antiplatelet therapy in thrombocytopenic patients is not well defined, and is based on expert opinion and small studies [102]. In this regard, the Society for Cardiovascular Angiography and Interventions (SCAI) suggests considering the use of thromboelastography (TEG), which offers a rapid and global evaluation of hemostasis, before PCI in oncologic patients with a platelet count < 30/10^9^L, in order to evaluate whether the procedure would be safe and possibly to guide hemostasis correction [103].

## 13. Gaps in Evidence

Currently, due to a lack of evidence, many concerns regarding the optimal management of patients with thrombocytopenia remain. First, there are no data that can exactly predict the specific risk of bleeding associated with different platelet count ranges in patients on antiplatelet therapy, making it difficult to estimate the net clinical benefit of antithrombotic treatment in these patients [104]. Moreover, it is still not clear if de-escalating DAPT or other antithrombotic approaches in thrombocytopenic patients may be reasonable in terms of thrombotic risk; indeed, while de-escalation therapy has been found to be safe and efficacious in high-bleeding-risk patients [104], no data are available for the subgroup of thrombocytopenic patients who are at high risk of thrombotic complications, and this subgroup should be investigated. Another relevant question regards patients requiring triple antithrombotic therapy, such as those with atrial fibrillation who undergo PCI, in whom basal bleeding risk is strongly increased and antiplatelet withdrawal may be associated with a significant increase in thrombotic recurrence [105]. Future research should fulfill these unmet needs, helping clinicians to manage these patients.

## 14. Conclusions

Antithrombotic therapy in patients with thrombocytopenia who are candidates for PCI represents a challenge, and should be guided by a careful evaluation of patients’ characteristics and history. A low platelet count influences bleeding risk, but does not protect against arterial thrombosis. Moreover, the assessment of individual bleeding risk may be complex, as bleeding tendency is a product of the interaction of many factors. More data are required to more effectively deal with thrombocytopenia during ACSs and to optimize thrombotic risk management, limiting bleeding occurrence.

## Figures and Tables

**Table 1 jcm-14-00838-t001:** Antithrombotic drugs used after PCI, their mechanisms of action and associated bleeding risks. LDA: low-dose aspirin.

Drug	Mechanism of Action	Bleeding Risk
Aspirin	Cicloxygenase-1 inhibition and thromboxane synthesis	1.7% major bleeding with LDA [21,22] 2.5% with >325 mg dose [22]
Ticlopidine	Inhibition of ADP P2Y12 receptor	2.1% major bleeding [22]
Clopidogrel	Inhibition of ADP P2Y12 receptor	2.1% major bleeding [22]
Prasugrel (on top of ASA)	Inhibition of ADP P2Y12 receptor	4.8% major bleeding (1 year) [11]
Ticagrelor (on top of ASA)	Inhibition of ADP P2Y12 receptor	5.2% major bleeding (1 year) [11]
Abciximab (on top of ASA)	Inhibition of α_IIb_β_3_ receptor	3.6% major bleeding [22]
Eptifibatide (on top of ASA)	Inhibition of α_IIb_β_3_ receptor	3.6% major bleeding [22]
Tirofiban (on top of ASA)	Inhibition of α_IIb_β_3_ receptor	3.6% major bleeding [22]

**Table 2 jcm-14-00838-t002:** Thrombocytopenia severity classification, according to platelet count [12].

Classification	Platelet Count
Grade 0	150–100 × 10^9^/L
Grade 1	100–75 × 10^9^/L
Grade 2	75–50 × 10^9^/L
Grade 3	50–25 × 10^9^/L
Grade 4	<25 × 10^9^/L

**Table 3 jcm-14-00838-t003:** Possible actions to mitigate bleeding risk in patients with ACS requiring revascularization.

Target	Action
Antithrombotic therapy	Prefer clopidogrel over new-generation P2Y12 inhibitors
Revascularization strategy	Consider plain old balloon angioplasty without stent placement in higher-risk patients
DAPT duration	De-escalate P2Y12 inhibitor within up to month
Type of stent	Prefer new generation stents to lower DAPT duration
Thrombocitopenia	Raise platelet count in accordance with thrombocytopenia pathogenesis to allow a full-dose antithrombotic approach

Table reports potential strategies to reduce risk of bleeding in patients with thrombocytopenia. DAPT: dual antiplatelet therapy.

**Table 4 jcm-14-00838-t004:** Approach for patients with thrombocytopenia and ACS, according to platelet count.

Platelet Count	Patients with ACS ^#^
100–150 × 10^9^	-Perform PCI with radial access followed ny conventional DAPT -Prefere new generation stent -Reduce DAPT duration up to 1 month if possible -De-escalate to clopidogrel as P2Y12 inhibitor if possible
50–100 × 10^9^	-Perform PCI with radial access followed by DAPT with clopidogrel as P2Y12 inhibitor -Prefere new generation stent -Reduce DAPT duration up to 1 month if possible
<50 × 10^9^	-Administer SAPT only if count > 10 × 10^9^ -Consider angioplasty without stent placement -Consider to rise platelet count > 50 × 10^9^ to allow DAPT and PCI in high thrombotic risk *

Table describes approach for patients with ACS and thrombocytopenia, according to platelet count. ^#^ In all thrombocytopenic patients, consider avoiding non-steroidal anti-inflammatory drugs and preventing gastroduodenal bleedings with proton pump inhibitors. * High-thrombotic-risk conditions include the following: STEMI, early symptom onset, life-threatening arrhythmias or hemodynamic instability, proximal coronary stenosis, multivessel disease. ACS: acute coronary syndrome; DAPT: dual antiplatelet therapy; PCI: percutaneous coronary intervention.

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
