# Peer review of "Antiplatelet Therapy in Low-Platelet-Count Patients After Percutaneous Coronary Intervention for Acute Coronary Syndromes"

_jcm, 2025, doi:10.3390/jcm14030838_

Round 1
Reviewer 1 Report
Comments and Suggestions for Authors
I congratulate the authors for a well-written article on an important and difficult clinical problem. Personally, I am more of a supporter of systematic literature reviews, but the format of a review article adopted by the authors is also acceptable. I suggest reviewing the entire text for minor errors - I noticed one typographical error in Figure 1.
Author Response
Comment: I congratulate the authors for a well-written article on an important and difficult clinical problem. Personally, I am more of a supporter of systematic literature reviews, but the format of a review article adopted by the authors is also acceptable. I suggest reviewing the entire text for minor errors - I noticed one typographical error in Figure 1.
Response: We really thank the referee for the kind comments. We carefully reviewed the text for errors. In order to comply with other referees’ comments we have now removed figure 1 from the manuscript.
Reviewer 2 Report
Comments and Suggestions for Authors
Authors reviewed the available evidence and the key points of the management of patients with a low platelet count undergoing coronary revascularization.
Although this manuscript is potentially interesting, several issues arise.
1. Antiphospholipid syndrome should be discussed as the cause of thrombosis with thrombocytopenia.
2. Monitoring for hemostasis may be helpful.
3. Simple table may be adequate instead of Figure 1.
4. Table which shows the algorism of antiplatelet therapy in after PCI with thrombocytopenia, may be helpful.
5. The relationship between age and antiplatelet therapy may be helpful.
Author Response
Comment 1: Antiphospholipid syndrome should be discussed as the cause of thrombosis with thrombocytopenia.
Response1 : We thank the referee for this constructive comment. Accordingly we have added a paragraph on antiphospholipid syndrome (lines 252-275).
Comment 2: Monitoring for hemostasis may be helpful.
Response 2: We appreciate this useful advice. Accordingly we have discussed this topic (lines 369-382).
Comment 3 Simple table may be adequate instead of Figure 1.
Response 3: According with this advice we substituted figure 1 with table 4.
Comment 4: Table which shows the algorism of antiplatelet therapy in after PCI with thrombocytopenia, may be helpful.
Response 4 We agree with this advice. We added the new table 5 summarizing the possible actions for the management of patients with thrombocytopenia undergoing PCI
Comment 5 The relationship between age and antiplatelet therapy may be helpful.
Response 5: We appreciate this comment. Accordingly we discussed the role of age in the stratification of bleeding risk as further element to be considered to evaluate antithrombotic therapy (lines 125-139).
Reviewer 3 Report
Comments and Suggestions for Authors
The authors of the manuscript have attempted to conduct a literature review on antiplatelet treatment options for patients who have experienced acute coronary syndrome (ACS), undergone percutaneous coronary intervention (PCI), and require combination antiplatelet therapy but have a low platelet count. The manuscript is topical and clinically significant.
Dual antiplatelet therapy (DAPT) is essential for secondary prevention of stent thrombosis in post-PCI patients, ideally for 12 months following ACS. However, in patients with thrombocytopenia, this treatment poses a higher risk of bleeding. Conversely, early de-escalation or cessation of DAPT is associated with a substantial risk of acute stent thrombosis, which carries high mortality rates.
I have the following comments and recommendations about the manuscript:
-
Abstract: I recommend the authors remove the first sentence of the abstract: "The risk of cardiovascular events increases strikingly after an acute coronary syndrome (ACS), particularly in the first months."
-
Citations: Some sentences lack citation indexes, which need to be added (e.g., lines 21–23).
-
Therapeutic clarifications are necessary at some parts of the text:
- Lines 29–30: The authors should specify that six months of dual antiplatelet therapy is recommended for patients with chronic coronary syndromes (CCS) after PCI.
- For ACS patients, the recommended duration is 12 months, extending up to 2 years in some cases using a reduced dose of ticagrelor, particularly in post-anterior STEMI patients with high or very high residual ischemic risk and low bleeding risk (as demonstrated by the first 12 months of treatment).
- Early de-escalation of dual therapy in both ACS and CCS should be reserved only for patients with high or very high bleeding risk and low thrombotic risk.
-
Tables:
- A legend for abbreviations is missing in the footer of Tables 1 and 3.
- Technical errors such as double brackets in lines 230 and 246 (e.g., "((61))") should be corrected.
-
Additional Section: I recommend adding a short section titled "Gaps in Evidence" at the end of the manuscript. This section should address unresolved issues in this area and highlight topics that require further clinical investigation.
Author Response
Comment 1: Abstract: I recommend the authors remove the first sentence of the abstract: "The risk of cardiovascular events increases strikingly after an acute coronary syndrome (ACS), particularly in the first months."
Response 1:We thank the referee for this advice, but we found that starting the abstract with the sentence “Dual antiplatelet therapy represents the mainstay of secondary prevention during this period” would make hardly understandable the premises of our work, thus we decided to maintain the first sentence.
Comment 2: Citations: Some sentences lack citation indexes, which need to be added (e.g., lines 21–23).
Response 2: The text has been reviewed for citations and these have been added
Comment 3: Therapeutic clarifications are necessary at some parts of the text:
-
- Lines 29–30: The authors should specify that six months of dual antiplatelet therapy is recommended for patients with chronic coronary syndromes (CCS) after PCI.
- For ACS patients, the recommended duration is 12 months, extending up to 2 years in some cases using a reduced dose of ticagrelor, particularly in post-anterior STEMI patients with high or very high residual ischemic risk and low bleeding risk (as demonstrated by the first 12 months of treatment).
- Early de-escalation of dual therapy in both ACS and CCS should be reserved only for patients with high or very high bleeding risk and low thrombotic risk.
Response 3: We really thank the referee for these relevant advices. We added these important informations.
Comment 4: Tables: A legend for abbreviations is missing in the footer of Tables 1 and 3.
Response to comment 4: The table legend has been added, as requested.
Comment 5: Technical errors such as double brackets in lines 230 and 246 (e.g., "(61)") should be corrected.
Response to comment 5: We carefully reviewed the text for typos.
Comment 6: Additional Section: I recommend adding a short section titled "Gaps in Evidence" at the end of the manuscript. This section should address unresolved issues in this area and highlight topics that require further clinical investigation.
Response 6: We thank the referee for this constructive comment. This section has been added lines (400-415).